# What Do Coffee Shop Entrepreneurs Need to Do to Raise Pro-Environmental Customer Behavioral Intentions?

**Sunmi Yun and Taeuk Kim ***

International Center for Hospitality Research & Development, Dedman School of Hospitality, Florida State University, Tallahassee, FL 32306, USA; syun@fsu.edu

*   Correspondence: tkim6@fsu.edu; Tel.: +1-850-345-2971

**Abstract:** Our research framework, built on the norm activation model (NAM), was designed to provide a comprehensive understanding of the formation of consumers' pro-environmental behavioral intentions in an eco-friendly coffee shop. We employed the NAM to test its mediating effect of personal environmental norms (PEN), social environmental norms (SEN), and ascription of responsibility (AR) and the moderating effect of the overall green image (OGI) on pro-environmental behavioral intentions. Data were collected through a survey of 530 customers who frequently visited a coffee shop in Korea, and structural equation modeling (SEM) was used to test the research hypotheses. The findings generally supported the hypothesized associations of the study variables within our proposed theoretical framework (PEN, SEN, and AR in order of the mediating effect on pro-environmental behavioral intentions) and confirmed OGI's moderating effect. In addition, the study's results have important (1) theoretical and (2) practical implications for the environment. (1) They expand the original NAM by explaining the effect of the relationship between SEN and PEN on pro-environmental customer behavioral intentions (PCBI) and confirm the mediating effect of the NAM (SEN, PEN, AR) on PCBI, as demonstrated in previous studies. (2) Moreover, the findings herein may encourage coffee shops to participate in the prevention of environmental problems by restricting the use of products such as plastic coffee cups and straws.

**Keywords:** norm activation (NAM); environmental problem concern (EPC); social environmental norms (SEN); personal environmental norms (PEN); ascription of responsibility (AR); pro-environmental customer behavioral intentions (PCBI); overall green image (OGI); coffee shops

## 1. Introduction

The global environment has become a primary focus of concern as more people are aware of sustainability-related issues than ever before. Reflecting the awareness of ecological issues that have been steadily increasing over the last two decades, the general public is now beginning to comprehend the impact these issues will have on their lives [1–4]. As consumers have recognized the importance of a sustainable green environment, environmental protection has become an essential issue in the consumer market [5]. Governments all over the globe are executing environmental policies as environmental protection is gathering global interest. Over fifty nations participated in the ocean plastics reducing campaign. For instance, India prohibited the use of disposable plastic material. Chile, Peru prohibited plastic bags. Nigeria set up waste recycling factories. Private organizations such as Internal Olympic Committee (IOC), World Wildlife Fund (WWF), LG electronics, and Volvo also joined plastic regulation campaigns. Additionally, global citizens are showing their interest via SNS, expressing their opinion about disposable plastic materials' rejection in the form of a viral

campaign [6]. Environmentally-conscious consumers recognize that our society faces a severe crisis due to environmental problems (e.g., climate change, water scarcity, and air pollution). Customers with environmental concerns prefer to buy and use pro-environmental goods and services, even if they are less convenient than the alternatives [7,8]. Therefore, it is necessary to understand the needs of eco-friendly consumers and strive for pro-environmental management in the consumer market by, for example, implementing environmentally-friendly technologies, initiating diverse eco-friendly programs, encouraging eco-friendly practices among customers and employees, modifying operation processes, and developing environmental policies and guidelines [2,7,9–11]. Notably, the global growth of the coffee shop industry has inevitably increased energy and resource consumption, from the amount of coffee that is harvested to the electricity needed to switch on the lights in the actual shops. This increased consumption correlates with a rise in the amount of waste produced, including packaging materials and coffee grounds [12]. The issue of environmental sustainability in the coffee shop industry is frequently discussed in the mainstream media in terms of disposable waste, often focused on coffee cups and grounds [13,14]. Therefore, as coffee shops attempt to become environmentally friendly in order to respond to consumers' enhanced awareness, an increasing number of coffee shops are proactively changing their operations by focusing on more eco-friendly practices. For example, Starbucks is working to shrink its environmental footprint and meet its customers' expectations by increasing the use of recycled cups, reducing waste, and conserving energy/water [15]. Dunkin' Donuts announced its plan to eliminate all polystyrene foam cups in its global supply chain beginning in spring 2018, with a targeted completion date of 2020 [16]. Costa Coffee has put great effort into using sustainable and recyclable products and equipment. For instance, the company's paper cups are made of a sustainable wood pulp material from northern European forests; the company does not use smart coffee machines in order to help reduce carbon emissions, and it recycles all of its coffee sacks into the shops' carpet underlay [16]. Despite this growing interest in pro-environmental behaviors, however, relatively little attention has been paid to the identification of the essential factors that influence pro-environmental customer behavioral intentions (PCBI) in environmentally-friendly coffee shops. PCBI can reduce severe environmental pollution and enable customers to improve their quality of life in a clean environment and move toward a sustainable future [2,3,17–19]. PCBI also improves business in coffee shops by increasing customer revisits due to the higher demand for eco-friendly goods and enhanced customer satisfaction and employee commitment [20]. Therefore, this study aims to expand the research on PCBI and benefit both the environment and coffee shops' ability to conduct business successfully.

Previous studies have relied heavily on Ajzen's [21] theory of planned behavior (TPB), Stern's [22] value-belief-norm (VBN) theory, and Schwartz's [23] norm activation model (NAM) as the most important predictors of PCBI within the domain of environmental psychology [3,24–26]. Of these, the NAM is considered the most influential theory [3,24,27,28]. The NAM used in this study is based on the NAM that researchers [3,27] developed by extending the VBN theory. The NAM implies the decision-making process of activating norms for pro-social behavior [3,29]. Schwartz's [23] NAM comprises three concepts: awareness of problems, ascription of responsibility (AR), and personal norms [3,22,23,30,31]. Although the NAM has been extensively used in PCBI [30], many researchers asserted the need to expand the original NAM to better explain individuals' eco-friendly intentions/behaviors [1,3,12,22,32–35]. In this study, the critical variable of the existing NAM is personal norms. In Schwartz's [23] study, personal norms are defined as "internalized rules of conduct that are socially learned [and that] vary among individuals within the same society and direct behavior in a particular situation". Hence, personal norms are behaviors that appear differently depending on individual tendencies and situations based on behaviors learned in society. Social norms affect individuals, and then personal norms appear. Therefore, unlike the previous studies mentioned above, social environmental norms (SEN) are applied to the NAM, and social norms influence personal norms and identify relationships with PCBI. Besides, previous studies on marketing and consumer behavior have indicated that the overall image of a firm plays a critical role in customers' behavioral

intentions [36]. References [5,37,38] found that the higher the overall green image (OGI), the more people behave in an eco-friendly manner.

To fill the existing research gaps, this study's purpose is to develop a theoretical framework that clearly explains PCBI in the environmentally-responsible coffee shop business, which has not yet been well documented. Moreover, we want to use customers' environmental awareness as a basis for marketing to improve the coffee shop business by identifying the role of personal and social norms to explain PCBI. Specifically, we aim to (1) broaden the NAM by incorporating the VBN framework and normative process; (2) test the mediating impact of personal environmental norms (PEN, SEN, and AR); (3) deepen the NAM by considering the moderating impact of OGI; (4) identify the adequacy of the proposed model by conducting a model comparison; and (5) examine the relative importance among constructs in building intentions in the model.

## 2. Conceptual Framework

### 2.1. Environmental Problem Concern

Protecting the environment is paramount, and increasing concern about environmental protection may facilitate proper discussion about this issue. Thus, various studies have been conducted to define the true nature of environmental concern and find its precise measurement methodology [35]. Environmental problem concern (EPC) can be classified as a user-defined term. However, it normally refers to "awareness about humans' capability to damage nature". EPC can induce people to protect nature through three phases: seeing, feeling, and acting [39]. Mostafa [40] insisted that environmental concern can positively influence consumers' intent to choose eco-friendly products. Evidence is offered in the results of numerous studies aimed at shedding light on the relationship between environmental concern and pro-environmental intentions/behaviors [41,42].

### 2.2. Norm Activation Model

Norm activation model (NAM) can be defined as "the standard of value judgment that a person should follow when that person thinks and acts". Amendments to people's thoughts and behaviors are made based on two types of norms: personal norms function from within and social norms that are constructed in the surrounding society [43]. In 1977, Schwartz developed the NAM in the context of altruistic behavior. The model was designed to investigate the true nature of altruistic behavior and to examine people's pro-environmental behaviors and intentions [34] in order to explain moral norm/pro-environmental action transfer [23]. Hopper and Nielsen [44] applied the NAM to recycling behavior, arguing that recycling should be interpreted as an altruistic behavior derived from norms. The researchers argued that recycling behavior could be described as altruistic behavior since sorting/moving recyclable goods does not bring anything to oneself but only benefits future society [44].

The NAM has played a critical role in research on pro-social intentions, and it contains three cardinal variables: environmental problem concern (EPC), personal environmental norms (PEN), and ascription of responsibility (AR) [23,45]. Specifically, PEN is the NAM's core element, described as a "moral obligation to perform or refrain from specific actions" [46]. PEN works as an index of pro-social behavior. EPC can be defined as "whether someone is aware of the negative consequences for others or for other things one values when not acting pro-socially" [45]. AR refers to "feelings of responsibility for the negative consequences of not acting pro-socially" [45]. For example, wasting electricity may lead to a number of negative consequences. However, people who have EPC may develop AR to feel joint responsibility for its negative consequences. Inversely, the development of AR is difficult if EPC is not yet developed. Studies have found that EPC positively affects the development of AR [3,34]. According to studies performed using empirical methods, EPC, AR, and PEN play an essential role in individuals' environmental intentions/behaviors [3,28,45,47–49]. Additionally, De Groot and Steg [41] found that the cooperation between EPC and AR positively affected people's

acceptance of several energy policies. Moreover, Guagnano [50] reported that the cooperation between EPC and AR leads people to buy recycled paper products. From this perspective, we can suggest that the cooperation of EPC and AR may boost people's AR for energy saving and develop further research on the relationship between EPC and AR. According to Han's study, evidence has also been produced through the empirical method, given the factors suggesting that EPC can affect AR, which can influence PEN to promote pro-environmental customer behavioral intentions (PCBI) [3]. Continuing the research stream on the relationships among variables within the norm activation model framework, this study proposes that EPC influences AR and PEN and that AR, in turn, affects PEN and PCBI.

**Hypothesis 1 (H1).** *Environmental problem concern is positively related to ascription of responsibility.*

**Hypothesis 2 (H2).** *Environmental problem concern is positively related to personal environmental norm.*

**Hypothesis 3 (H3).** *Ascription of responsibility is positively related to personal environmental norm.*

**Hypothesis 4 (H4).** *Ascription of responsibility is positively related to pro-environmental customer behavioral intentions.*

*2.3. Relationships among Environmental Problem Concern, Ascription of Responsibility, Personal Environmental Norms, and Pro-Environmental Customer Behavioral Intentions*

Environmental problem concern (EPC) refers to a sense of "knowing of the impact of human behavior on the environment" [50]. Since many environmental problems are slow-paced, complex, and indirectly related to individuals' lives, a cognitive limitation occurs that makes it difficult for people to recognize these problems. People's indifference to and emotional reactions against environmental problems work as emotional limitations [51]. Studies have found that the solution to this is related to the positive effect of pro-environmental behaviors/intentions on EPC [52]. For instance, people with increased EPC prefer to purchase eco-friendly products or organic foods and tend to participate in recycling programs [53]. Some people stop using hairspray after learning that chlorofluorocarbon damages the ozone. In association with the eco-friendly hotel management system, Chan and Hawkins [54] indicated that increased EPC and understanding of the system might lead to eco-friendly acts.

A belief that environmental problems will threaten individual values can activate pro-environmental actions derived from personal environmental norm (PEN). In this case, PEN creates a general predisposition that influences all kinds of pro-environmental behaviors/intentions [55]. For example, it is known that personal energy waste can create some negative consequences in the long-term (e.g., power failure, environmental damage, and even global warming). In this case, EPC can lead individuals who might be concerned to actively approach energy saving [56]. Prior research also reported that EPC has a significant influence on PEN. For instance, De Groot and Steg [45] found that respondents with high EPC had higher PEN than respondents with low EPC. Harland et al. [57] also found that high EPC can positively influence PEN in normal households, inducing the usage of public transportation and water conservation. Similarly, high EPC can positively influence a hotel employee's PEN to save energy [56].

In the norm activation model (NAM), EPC and ascription of responsibility (AR) are emphasized as key factors that promote one's personal obligation to engage in altruistic behavior. However, it is also true that AR was defined as a moderator between PEN and pro-environmental behaviors/intentions [46,58]. Scholars have found that the relationship between AR and PEN can be positive [47]. For example, Bamberg et al. [58] showed that AR exerts a strong positive effect on PEN in the context of car use. Han et al. [59] found a positive influence of AR on personal norms when travelers intended to stay at an environmentally responsible lodging. Stern's [22] value-belief-norm (VAN) theory was an attempt to link the NAM to the relationship between general values and pro-environmental intentions/behaviors. The VAN is thereby also an integrative theory in itself and it assumes that

pro-environmental intentions/behaviors can be directly determined by PEN, which is based on the NAM. In addition, Stern assumed that PEN has to be activated by AR and EPC. However, he also assumed EPC as a prerequisite of AR. According to the VBN theory, EPC is related to one's general ecological worldview, which is measured by the new environmental paradigm [60]. In the proposed research model, PEN is activated by EPC and AR; moreover, EPC is posited to be an antecedent of AR. People with AR are likely to feel responsible for their acts and be less likely to perform certain behaviors [34]. A person's intention to choose organic food is formulated by attitude, subjective norms, perceived behavioral control, and PEN [61,62]. Privileging organic food, in turn, promotes visiting restaurants that feature organic menu items [34].

Ultimately, PEN is activated by a certain level of EPC and people's feeling of responsibility to alleviate the problem. When people understand the severity of an environmental problem, they are more likely to undertake pro-environmental behavior [29]. For instance, people who are aware of the relationship between car usage and air pollution develop a sense of responsibility for alleviating air pollution, which leads them to reduce their car use [63]. Likewise, the NAM has been successfully used to explain many kinds of pro-environmental behaviors, such as willingness to choose organic food [64], intentions to revisit an eco-cruise [3,7], intentions to reduce energy use [59,65], and recycling behaviors [44,66–68]. According to Schwartz [23], behaviors reflect a person's internalized value system, and certain behaviors are performed when the relevant values and norms are about to be activated. The NAM posits that people with EPC and AR for their activities are more likely to display altruistic behavior. When EPC and AR are present, people tend to act in ways that benefit others [69]. Accordingly, we hypothesize:

**Hypothesis 5 (H5).** *Personal environmental norm is positively related to pro-environmental customer behavioral intentions.*

*2.4. Relationships among Environmental Problem Concern, Ascription of Responsibility, Social Environmental Norms, and Pro-Environmental Customer Behavioral Intentions*

Environmental problem concern (EPC) is a concept interest in environmental problems and/or consequences [70]. Social environmental norm (SEN) can be compared to ethos, an air that induces people to have certain environmental problem concern (EPC). SEN is mentioned as an essential concept that helps explain individuals' pro-environmental customer behavioral intentions (PCBI) [2,3,9,67,69,70]. Moreover, in the examination of environmentally-friendly tourism behaviors, SEN has been used as an index of individual behavior [32,47,68–75]. The norm activation model (NAM) proposes that SEN can influence an individual's actual behaviors through its influence on EPC [23]. In conjunction with self-concept, personal environmental norm (PEN) is expressed as a moral obligation to perform a certain behavior [23,76]. It can be interpreted as "a longing to do certain acts" and a behavior motivated by a will to act according to one's values [77]. In this case, SEN promotes a compulsion derived from society's current ethos to engage in certain acts, like environmental protection. For example, a tidy and organized space creates an ethos in itself that encourages people to keep that place unsoiled. The role of injunctive social norms is to suggest to people that they must follow specific social rules or there would be a sanction [78]. A person who visits a tourist site may avoid littering for these reasons.

Experimental studies have shown that SEN can have powerful effects on willingness to engage in pro-environmental behavior [43,79], and a significant relationship between SEN and PEN and their impact on environmentally-friendly purchasing intentions have been demonstrated in the context of hospitality [3]. According to the NAM, SEN and PEN are interrelated; SEN influences PEN to modify individuals' actual behaviors [23]. On the other hand, personal norms are defined as individuals' own beliefs linked to their self-concepts. The NAM suggests various ways to understand SEN's effect on pro-environmental intentions/behaviors [30,78,80]. The NAM was widely used to shed light on "a decision-making process through which personal and social norms mediate the influences of general values on altruistic and/or helping behavior" [46] in the context of pro-environmental behavior [41].

However, ever since the potential of the NAM's extensive use in the environmental domain was uncovered [81], behaviors such as the burning of garden waste [60], recycling [44,49,67], and energy conservation [82] have been studied using the NAM.

Most studies interpret the NAM using either a mediation model or a moderation model. The mediation model assumes that EPC influences PEN through ascription of responsibility (AR) [33,45]. In the moderation model, PEN's influence on behavior is moderated by EPC and AR. In this study, we interpret the NAM as a mediation model, as De Groot and Steg [45] provided strong evidence of the NAM as a mediation model via five recent studies comparing the two models.

Researchers proposing a mediation model assume that PEN and AR have indirect effects on intentions and behaviors via PEN [66,83,84]. More specifically, PEN is assumed to mediate the relationship between AR and pro-social intentions and behaviors, and AR is assumed to mediate the relationship between EPC and PEN. This interpretation of the NAM has been supported by several studies [45]. Stern et al. [85] showed that PEN can be predicted by the level of AR. It has also been shown that PEN can be a significant predictor of several pro-environmental behaviors. The application of the NAM enabled the identification of PEN as a mediator of behaviors' situational factors [29], and environmental studies found influential factors of pro-environmental behaviors. For instance, Vining and Ebreo [49] found that EPC influenced recycling behavior even when PEN's central role was not yet identified. Moreover, some researchers suggest that PEN mediates the relationship between all of the NAM's independent variable components including EPC, AR, and PEN. In addition, SEN is generally believed to be an effective predictor of PEN and PCBI [2,3,7,10,47]. According to Kim et al. [86], SEN effectively mediates the relationship between green identity and customers' purchase intentions. Ultimately, it can be assumed that EPC influences SEN while SEN affects PCBI and PEN within our conceptual framework. Hence, the following hypotheses were formulated:

**Hypothesis 6 (H6).** *Environmental problem concern is positively related to social environmental norm.*

**Hypothesis 7 (H7).** *Social environmental norm is positively related to pro-environmental customer behavioral intentions.*

**Hypothesis 8 (H8).** *Social environmental norm is positively related to personal environmental norm.*

**Hypothesis 9 (H9).** *Ascription of responsibility, social environmental norm, and personal environmental norm significantly mediate the relationship between environmental problem concern and pro-environmental customer behavioral intentions.*

*2.5. Moderating Effect of Overall Green Image*

According to Martineau [87], "store image" is the reflection of a shopper's awareness about a store, partly based on function and partly based on its atmosphere made up of psychological attributes. Subsequent studies have focused on the relationship between store image and customers' perceptions. Moreover, brand image creates brand awareness, which prompts consumers to make purchase decisions [88]. Ur and Ishaq [89] proved that store image has a significant moderating effect on the relationship between brand image and purchase intention. Furthermore, store image became known as a manipulator of consumers' price assessments, perceptions, satisfaction, and intentions/behaviors. Therefore, store image can be treated as a critical concept in consumer behavior research [36]. In addition, Bloemer and De Ruyter [36] have insisted that store image is a firm's most important attribute, alluding to consumers' core element of overall perceptions of a firm. According to Keller [90], "firm image" is "a series of perceptions about a firm as reflected by its associations in consumers' memories".

Customers' perceptions of a store's overall image can be converted to overall green image (OGI) [36]. Wang et al. [37] evaluated whether customers perceived a restaurant's green image as its green practices. OGI, which refers to the tendency to perform eco-friendly practices, can be used in

brand advertising since the consumers' minds can personally link the green image with environmental commitments and concerns [91]. Customers in the hotel industry with OGI are more likely to engage in eco-friendly practices [38]. Similarly, a restaurant's OGI can relate consumers' perceptions of the restaurant with pro-environmental customer behavioral intentions (PCBI).

In consumer behavior, scholars have stressed the role of image for firms and products. Jeong et al. [38] tested the impacts of eco-friendly practices on a cafe's green image and its customers' attitudes. The results showed that a green image fostered positive customer attitudes toward the café, eventually affecting consumption decisions. Han [5] suggested that enhancing a hotel's green image can manipulate customers' pro-environmental intentions, which affects consumption decisions. Thus, this strategy can significantly benefit hoteliers in the green hotel industry. Based on these research findings, it can be assumed that the strength of the relationship between ascription of responsibility (AR), personal environmental norm (PEN), social environmental norm (SEN), and pro-environmental customer behavioral intentions (PCBI) depends on the level of overall green image. Based on the discussion of the relationship between SEN, PEN, AR, and PCBI, three hypotheses are proposed as follows:

**Hypothesis 10a (H10a).** *Overall green image has a significant moderating role in the relationship between ascription of responsibility and pro-environmental customer behavioral intentions.*

**Hypothesis 10b (H10b).** *Overall green image has a significant moderating role in the relationship between personal environmental norm and pro-environmental customer behavioral intentions.*

**Hypothesis 10c (H10c).** *Overall green image has a significant moderating role in the relationship between social environmental norm and pro-environmental customer behavioral intentions.*

## 3. Materials and Methods

### 3.1. Measures and Questionnaire

The study's online questionnaire included two sections. The first part of the survey requested demographic information (e.g., gender, marital status, age, educational level, annual income level, purpose of visits, and number of visits). The second part of the conceptual model constructs environmental problem concern (EPC), ascription of responsibility (AR), personal environmental norm (PEN), social environmental norm (SEN), overall green image (OGI), and pro-environmental customer behavioral intentions (PCBI). The measurement scales can be found in the literature [2,3,10,22,23,29–31,34]. Specifically, EPC featured three items (e.g., "I try not to buy from a brand of coffee shop that strongly pollutes") adopted from Choi et al. [1]; AR featured four items (e.g., "I feel jointly responsible for the exhaustion of energy sources") obtained from Steg and De Groot [31]; PEN featured three items (e.g., "I feel an obligation to visit environmentally responsible coffee shops rather than regular coffee shops") adopted from Berenguer [24]; SEN featured three items (e.g., "Most people who are important to me would believe that I visit an environmentally responsible coffee shop") obtained from Han et al. [5]; OGI featured three items (e.g., "My overall impression regarding a pro-environmental coffee shop is important") adopted from Han, Hsu, and Lee [5]; and, finally, PCBI featured eight items (e.g., "To protect the environment, I would expend effort to visit an environmentally responsible coffee shop rather than a general coffee shop in the future") obtained from Han et al.[3]. The measurement items used in this study are presented in the Appendix A. A seven-point Likert scale was used for all the measurement items in this research, ranging from "Extremely disagree" (1) to "Extremely agree" (7). In sum, we used 24 items for the assessment of the six variables in this study.

### 3.2. Data Collection and the Sample

The questionnaire was conducted with general coffee shop customers in Seoul, Republic of Korea via an online, self-administered questionnaire between 4 and 24 February 2019. In order to reduce the

common method bias (CMB) [92,93], procedural remedies were used in this study. First, in order to improve scale items before participants received the questionnaire, we supplemented the ambiguous expressions and questions under the guidance of five professors affiliated to Department of Foodservice Management from 21–24 January 2019. In addition, to reduce the CMB, the questionnaire was designed to protect respondent anonymity and reduce evaluation apprehension. Moreover, the question order was counterbalanced. Survey distribution and collection were performed with the help of Embrain, an online survey company that provides reliable access to a large participant pool of randomly selected, voluntary participants from national consumer panel groups. In addition, our survey questionnaire was distributed to various coffee shop customers in Seoul, Republic of Korea who had visited a coffee shop in the month prior to the survey. After the exclusion of insincere and inappropriate responses, a total of 530 usable responses were collected. These cases were used to evaluate the adequacy and test the hypothesized relationships of the proposed theoretical framework.

Regarding the demographic characteristics of the 530 participants, 57.9% ($n = 307$) were female customers and 42.1% ($n = 223$) were male customers. Moreover, approximately 55.7% ($n = 295$) of the respondents were single while 37.9% were married ($n = 201$), 4.2% were divorced ($n = 22$), 0.9% were separated ($n = 5$), and 2.1% were widowed ($n = 7$). In addition, 79.4% ($n = 421$) of respondents were 20–40 years old, and the remaining 20.6% ($n = 109$) were over 40 years old. Their levels of education were 2-year college (43.6%, $n = 231$), 4-year college (33.4%, $n = 231$), high school graduate and below (18.7%, $n = 99$), and postgraduate or higher (4.4%, $n = 23$). Most of the respondents had an annual income of 20,000,000–29,999,999 won (35.1%, $n = 186$), followed by 20,000,000 won or below (24.7%, $n = 24.7$), 30,000,000–39,999,999 won (23.8%, $n = 126$), and 40,000,000 won and over (17.4%, $n = 87$). The most common responses regarding the purpose of coffee shop visits were studying (38.9%, $n = 206$), meeting friends (28.3%, $n = 150$), business meetings (23.2%, $n = 123$), and private time (9.6%, $n = 51$). The numbers of visits to coffee shops per week were 1–2 times (44.2%, $n = 234$), 3–4 times (39.8%, $n = 211$), 5–6 times (13.2%, $n = 70$), and 6+ times (2.9%, $n = 15$).

### 3.3. Data Analysis and Tools

To analyze the data, this study used IBM SPSS 20.0 program (SPSS Inc., Chicago, IL, USA) and AMOS 24.0 program (SPSS Inc., Chicago, IL, USA). First of all, demographic characteristics and correlation were used by IBM SPSS 20.0 program. Furthermore, the measurement model with confirmatory factor analysis (CFA), construct structural equation modeling (SEM), chi-square difference test for modeling comparison, and metric invariance for moderating effect was evaluated by AMOS 24.0 program.

## 4. Results

### 4.1. Common Method Variance

This study intended to mitigate possible common method variance (CMV) by using two procedural remedies in the survey design and using the analysis method. First of all, this study used different cover stories when the respondents answered using the same scale to facilitate the psychological separation between criterion variables and predictors. For example, the cover story for the pro-environmental concern scale was "The following statements are irrelevant to the above questions. Please read carefully each statement and then mark from extremely disagree to agree with your recent feeling". Another method of reducing CMV was dealing with item ambiguity, as suggested by [94]. For example, this study provided specific definitions of imprecise terms to help respondents' comprehension. Moreover, the results of Harman's one-factor analysis, which is a post hoc test to detect possible CMV [94], showed that CMV of the unmeasured latent methods factor was 1.1% and the one-factor measurement model was confirmed to fit the data satisfactorily (goodness-of-fit statistics for the measurement model: $\chi^2 = 237.284$, df = 121, $p < 0.001$, $\chi^2$/df = 1.961, RMSEA = 0.043, CFI = 0.983, IFI = 0.983, TLI = 0.976).

This study made an effort to reduce CMV by employing procedural remedies in the survey design stage. Therefore, CMV did not influence the parameter estimations.

### 4.2. Reliability and Validity Assessments and Confirmatory Factor Analysis

The study's measurement model was generated by conducting a confirmatory factory analysis. We used the item parceling strategy (IPS) to increase the scale's reliability [95,96]. If the measurement index is composed of multiple items, the items are grouped into two, three, or more groups according to structural concepts, and then the respective items are summed or averaged to be used as indicators of the new item bundles (parcels) method [97]. Thus, to construct a balanced method, we used eight items in three factors based on the PCBI factor load derived from the items' confirmatory factor analysis. A maximum likelihood estimation approach was used for the analysis. The model was confirmed to fit the data satisfactorily (goodness-of-fit statistics for the measurement model: $\chi^2$ = 327.878, df = 137, $p$ <0.000, $\chi^2$/df = 2.393, RMSEA = 0.051, CFI = 0.972, IFI = 0.972, TLI = 0.965). The RMSEA should range between 0.04 and 0.08 [98] and the TLI, IFI, and CFI values should be close to 1.00, which indicates an acceptable fit [99]. The average variance extracted (AVE) value was then estimated. The calculation of the AVE revealed that all AVE values exceeded the minimum threshold of 0.50 [100] and they ranged from 0.928 to 0.968 (as displayed in Table 1). These values were greater than the square of the correlation between two variables. Therefore, discriminant validity was supported. The construct reliability (CR) for the six study constructs was also calculated. The values of CR ranged from 0.974 to 0.990 (as shown in Table 1). These values highly exceeded Bagozzi and Yi's [101] suggested cutoff of 0.70, thus supporting convergent validity.

**Table 1.** Descriptive statistics of the constructs and correlations.

| Construct and Scale Item | | Standardized Loading | MEAN (SD) | AVE (CR) | EPC | AR | PEN | SEN | OGI | PCBI |
|---|---|---|---|---|---|---|---|---|---|---|
| EPC | EPC1 | 0.797 | 3.888 (1.183) | 0.928 (0.974) | 1 | - | - | - | - | - |
| | EPC2 | 0.882 | | | | | | | | |
| | EPC3 | 0.773 | | | | | | | | |
| AR | AR1 | 0.873 | 4.159 (0.999) | 0.928 (0.974) | 0.657 *** (0.431) | 1 | - | - | - | - |
| | AR2 | 0.826 | | | | | | | | |
| | AR3 | 0.849 | | | | | | | | |
| | AR4 | 0.741 | | | | | | | | |
| PEN | PEN1 | 0.832 | 4.236 (1.016) | 0.940 (0.979) | 0.699 *** (0.489) | 0.770 *** (0.593) | 1 | - | - | - |
| | PEN2 | 0.791 | | | | | | | | |
| | PEN3 | 0.753 | | | | | | | | |
| SEN | SEN1 | 0.821 | 4.090 (1.005) | 0.935 (0.977) | 0.660 *** (0.435) | 0.628 *** (0.394) | 0.834 *** (0.696) | 1 | - | - |
| | SEN2 | 0.742 | | | | | | | | |
| | SEN3 | 0.786 | | | | | | | | |
| OGI | OGI1 | 0.781 | 4.045 (1.019) | 0.953 (0.984) | 0.495 *** (0.245) | 0.498 *** (0.248) | 0.613 *** (0.376) | 0.519 *** (0.269) | 1 | - |
| | OGI2 | 0.835 | | | | | | | | |
| | OGI3 | 0.889 | | | | | | | | |
| PCBI | WOMI | 0.890 | 4.077 (0.982) | 0.964 (0.988) | 0.576 *** (0.332) | 0.664 *** (0.441) | 0.739 *** (0.546) | 0.730 *** (0.633) | 0.755 *** (0.570) | 1 |
| | WTPI | 0.912 | | | | | | | | |
| | RI | 0.755 | | | | | | | | |

Note 1. EPC = environmental problem concern, AR = ascription of responsibility, PEN = personal environmental norm, SEN = social environmental norm, PCBI = pro-environmental customer behavioral intentions, OGI = overall green image, AVE = average variance extracted, AVE = average variance extracted, CR = composite reliability, SD = standard deviation. Note 2. Goodness-of-fit statistics for the measurement model: $\chi^2$ = 327.878, df = 137, $p$ < 0.001, $\chi^2$/df = 2.393, RMSEA = 0.051, CFI = 0.972, IFI = 0.972, TLI = 0.965. Note 3. All factor loadings are significant at *** $p$ < 0.001.

### 4.3. Research Hypotheses Testing and Structural Equation Modeling (SEM)

The structural equation model (SEM) was generated using the hypothesized interrelationships among variables and the maximum likelihood estimation method. Moreover, SEM analysis was performed by using the maximum likelihood method [99] as an estimation method for the evaluation of both the model and procedures. SEM was shown to fit the data adequately (goodness-of-fit statistics for the structural model: $\chi^2$ = 323.864, df = 96, $p$ < 0.001, $\chi^2$/df = 3.374, RMSEA = 0.069, CFI = 0.959,

IFI = 0.959, TLI = 0.949). Moreover, SEM showed high prediction power for pro-environmental customer behavioral intentions (PCBI) in general ($R^2$ = 0.599). The standardized path coefficients and t-values are shown in Table 2. In addition, the hypothesis test results are provided in Figure 1. The path estimates show that environmental problem concern (EPC) had a significantly positive direct effect on ascription of responsibility (AR) ($\beta$ = 0.721, $p < 0.001$); thus, H1 was supported. The result of estimation indicated that EPC had a not significant effect on personal environmental norm (PEN); thus, H2 was not supported. Moreover, H3 was not supported because EPC was not positively and significantly associated with PEN. The result of estimation indicated that AR had a significant positive effect on PCBI ($\beta$ = 0.273, $p < 0.001$); thus, H4 was supported. The impact of PEN on PCBI ($\beta$ = 0.384, $p < 0.001$) was assessed; thus, H5 was supported. The proposed impact of EPC on social environmental norm (SEN) was assessed. As expected, EPC had an impact on SEN ($\beta$ = 0.761, $p < 0.01$); thus, H6 was supported. The influence of SEN on PCBI was also evaluated. It was found that SEN had a significant positive effect on PCBI ($\beta$ = 0.230, $p < 0.01$); thus, H7 was supported. As expected, the link between SEN and PEN was significant ($\beta$ SEN-PEN = 0.384, $p < 0.01$); thus, H8 was supported. Moreover, the results of the analysis on mediating effects are shown by significant indirect effects in Table 3. This study used the bootstrapping by [102] to verify the significance of the mediation effect. The findings revealed that EPC significantly affected PCBI ($\beta$ EPC $\rightarrow$ AR & SEN & PEN $\rightarrow$ PCBI = 0.640, $p < 0.001$) indirectly through AR, SEN, and PEN, thus confirming all of them as partial mediating variables. Thus, H9 was supported.

**Table 2.** Structural model results and hypotheses testing.

| Hypothesized Paths | Coefficients | *t*-Values |
|---|---|---|
| H1: EPC $\rightarrow$ AR | 0.721 | 13.426 *** |
| H2: EPC $\rightarrow$ PEN | 0.128 | 1.492 |
| H3: AR $\rightarrow$ PEN | 0.038 | 1.765 |
| H4: AR $\rightarrow$ PCBI | 0.273 | 6.106 *** |
| H5: PEN $\rightarrow$ PCBI | 0.384 | 4.437 *** |
| H6: EPC $\rightarrow$ SEN | 0.761 | 15.109 *** |
| H7: SEN $\rightarrow$ PCBI | 0.230 | 2.652 ** |
| H8: SEN $\rightarrow$ PEN | 0.384 | 4.437 *** |
| **Explained variable:** | $R^2$(AR) = 0.520 | $R^2$(PEN) = 0.580 |
| | $R^2$(PCBI) = 0.599 | $R^2$(SEN) = 0.701 |

Note 1. EPC = environmental problem concern, AR = ascription of responsibility, PEN = personal environmental norm, SEN = social environmental norm, PCBI = pro-environmental customer behavioral intentions, AVE = average variance extracted, CR = composite reliability, SD = standard deviation. Note 2. Goodness-of-fit statistics for the structural model: $\chi^2$ = 323.864, df = 96, $\chi^2$/df = 3.374, RMSEA = 0.069, CFI = 0.959, IFI = 0.959, TLI = 0.949. ** $p < 0.01$, *** $p < 0.001$.

**Table 3.** Results of mediation effect ascription of responsibility, personal environmental norm, and social environmental norm.

| Indirect Effect: | $\beta$(EPC $\rightarrow$ AR & PEN & SEN $\rightarrow$ PCBI) = 0.640 *** |
|---|---|
| **Total Effect on PCBI:** | $\beta$(SEN) = 0.640 *** |
| **The Results:** | H9: supported |

Note 1. EPC = environmental problem concern, AR = ascription of responsibility, PEN = personal environmental norm, SEN = social environmental norm, PCBI = pro-environmental customer behavioral intentions, AVE = average variance extracted, CR = composite reliability, SD = standard deviation. *** $p < 0.001$.

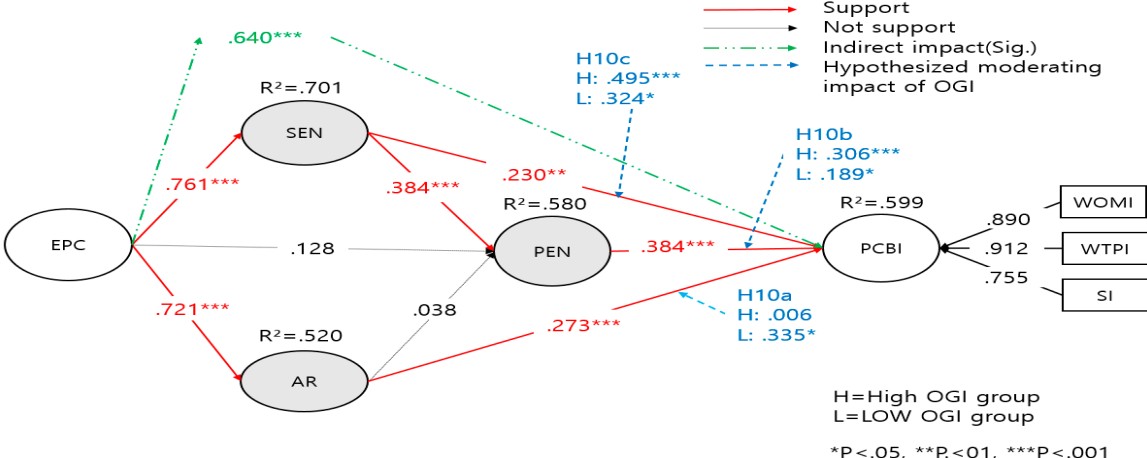

**Figure 1.** Structural equation model estimation and test for structural metric invariance. Note 1. EPC = environmental problem concern, AR = ascription of responsibility, PEN = personal environmental norms, SEN = social environmental norms, PCBI = pro-environmental customer behavioral intentions, OGI = overall green image, WOMI = word of mouth intention, WTPI = willingness to pay intention, SI = sacrifice intention. Note 2. Goodness-of-fit statistics for the measurement model: $\chi^2$ = 327.878, df = 137, $p < 0.001$, $\chi^2$/df = 2.393, RMSEA = 0.051, CFI = 0.972, IFI = 0.972, TLI = 0.965. Note 3. Goodness-of-fit statistics for the structural model: $\chi^2$ = 323.864, df = 96, $p < 0.001$, $\chi^2$/df = 3.374, RMSEA = 0.069, CFI = 0.959, IFI = 0.959, TLI = 0.949. Note 4. Goodness-of-fit statistics for the baseline model: $\chi^2$ = 409.868, df = 192, $p < 0.001$, $\chi^2$/df = 2.135, RMSEA = 0.046, CFI = 0.953, IFI = 0.954, TLI = 0.942. Note 5. Two identical structural models were evaluated (model for high- [$n$ = 322] and low- [$n$ = 207] OGI groups).

## 4.4. Moderating Effect of Overall Green Image

To assess the proposed moderating impact of overall green image (OGI), a test for metric invariance was conducted. In this case, a continuous variable—such as an interval scale or ratio scale—was used as a control variable to evaluate a metric variable. The sample was divided into high ($n$ = 322) and low ($n$ = 207) overall image groups through K-mean cluster analysis. Since the significance probability for the nested model was 0.152, which exceeds the significance level of 0.05, OGI was found to have an effect. Moreover, the baseline model showed an acceptable level for data suitability (goodness-of-fit statistics for the baseline model: $\chi^2$ = 409.868, df = 192, $p < 0.001$, $\chi^2$/df = 2.135, RMSEA = 0.046, CFI = 0.953, IFI = 0.954, TLI = 0.942). The details are shown in Table 4 and Figure 1. The chi-square test results with measurement invariance revealed that the path from AR to pro-environmental customer behavioral intentions (PCBI) for coffee shop practices was not significantly different between groups ($\Delta\chi^2$ = 0.199, $p > 0.05$). However, while the ascription of the responsibility (AR)–PCBI link for the low-OGI group ($\beta$ = 0.335, $p < 0.05$) was significant, the AR–PCBI path for the high-OGI group was not significant. Thus, the chi-square test results did not differ significantly across groups; thus, H10a was supported. The study's findings also indicated that, although the relationship between personal environmental norm (PEN) and PCBI differed significantly between the high ($\beta$ = 0.306, $p < 0.001$) and low ($\beta$ = 0.189, $p < 0.05$) OGI groups, there was no significant difference in the PEN–PCBI link ($\Delta\chi^2$ = 0.402, $p > 0.05$); thus, H10b was not supported. Moreover, the study uncovered that the SEN–PCBI link was significantly different between the high- ($\beta$ = 0.495, $p < 0.001$) and low ($\beta$ = 0.324, $p < 0.05$) OGI groups, but that there was no significant difference in the SEN–PCBI link ($\Delta\chi^2$ = 2.501, $p > 0.05$); thus, H10c was not supported.

**Table 4.** Results of the moderating effect of overall green image.

| Paths | High-OGI Group (*n* = 322) | | Low-OGI Group (*n* = 207) | | Baseline Model (Freely Estimated) | Nested Model (Constrained to Be Equal) |
|---|---|---|---|---|---|---|
| | Coefficients | *t*-Values | Coefficients | *t*-Values | | |
| H10a: AR–PCBI | 0.006 | 0.050 | 0.335 | 2.096 * | $\chi^2$ (192) = 409.868 | $\chi^2$ (193) = 410.067$^a$ |
| H10b: PEN–PCBI | 0.306 | 4.805 *** | 0.189 | 2.582 * | $\chi^2$ (192) = 409.868 | $\chi^2$ (193) = 410.270$^b$ |
| H10c: SEN–PCBI | 0.496 | 4.292 *** | 0.324 | 2.047 * | $\chi^2$ (192) = 409.868 | $\chi^2$ (193) = 412.369$^c$ |

| Chi-square difference test: | Test results: | Goodness-of-fit statistics for the baseline model: |
|---|---|---|
| $^a \Delta\chi^2$ (1) = 0.199, *p* > 0.05 | H10a: Supported | $\chi^2$ = 409.868, df = 192, *p* < 0.001, $\chi^2$/df = 2.135, |
| $^b \Delta\chi^2$ (1) = 0.402, *p* > 0.05 | H10b: Not supported | RMSEA = 0.046, CFI = 0.953, IFI = 0.954, TLI = 0.942. |
| $^c \Delta\chi^2$ (1) = 2.501, *p* > 0.05 | H10c: Not supported | * *p* < 0.05, *** *p* < 0.001 |

Note 1. AR = ascription of responsibility, PEN = personal environmental norms, SEN = social environmental norms, PCBI = pro-environmental customer behavioral intentions, OGI = overall green image.

## 5. Discussion and Implications

The study's proposed conceptual model was designed to clearly explain pro-environmental customer behavioral intentions (PCBI) of coffee shop customers. In the eco-friendly coffee shop context, there is limited scholarly research on PCBI for environmentally responsible practices during coffee shop visits. The proposed theoretical framework comprising coffee shop practices of environmental problem concern (EPC) as independent variables; ascription of responsibility (AR), personal environmental norms (PEN), and social environmental norms (SEN) as mediators; and overall green image (OGI) as a moderator was demonstrated to be useful and to satisfactorily predict PCBI. The hypothesized relationships among study constructs were generally supported, but PEN was not affected by EPC or AR. A significant mediating impact of PEN, SEN, and AR was also found. Moreover, the test for metric invariance demonstrated OGI's moderating role. This study's findings generally provided a study framework for research constructs and their relationships in the area of eco-friendly coffee shops and customers' pro-environmental behaviors. Thus, useful theoretical implications for research related to PCBI and the practical implications of PCBI are suggested below.

### 5.1. Theoretical Implications

This study has several theoretical implications. First, the research establishes a specific set of consumer pro-environmental psychographics (concern, responsibility, and norms) as clear predictors of PCBI, which expands the current literature and provides guidelines for a clearer understanding of PCBI. In this sense, for instance, it can be deduced that increased consumption of organic coffee and foods in the region can lead to a concurrent increase in the use of recycled coffee cups [15–17]. Second, the original NAM has been used for over 40 years; however, more recently, the expanded version of the norm activation model (NAM) [3,4,29,33] based on theory of planned behavior (TPB) and value-belief-norm (VBN) has become popular. This study builds upon the original NAM, showing EPC's influence on PEN and SEN and its correlated effect on PCBI. This means that the effect of the relationship between PEN and SEN on PCBI provides a basis for future research examining PCBI in relation to PEN and SEN. Third, the study's empirical results indicate that PEN has the strongest direct effect on PCBI. The importance of PEN in explaining individuals' pro-environmental intentions and behaviors has been demonstrated by various studies [10,27,48]. These findings suggested that PEN was a major factor in determining PCBI. Fourth, by verifying the mediating effects of the NAM (SEN, PEN, and AR), this study supports a number of previous studies [3,4,29,31,45,66,88] in contrast to other previous studies [3,46,57,62] that used moderating effects. Fifth, this study reveals OGI's moderating role in the effects of AR, PEN, and SEN on PCBI, legitimizing OGI and supporting its inclusion in the model of PCBI as an important situational factor. This study is the first to apply the OGI variable that affects the eco-friendly behavior of coffee shop customers to the NAM.

*5.2. Practical Implications*

This study offers a number of practical implications for improving PCBI that can provide insights to coffee shop managers. First of all, the study's findings can inform coffee shops managers that the enhancement of EPC has a significant impact on AR and PCBI but no effect on PEN by pro-environmental decision-making. These results are different from those of the extended NAM. In the extended NAM, EPC, AR, and PEN all play an essential role in PCBI [4,28,34,45,47–50]. EPC and AR are highlighted as key factors that promote one's personal obligation to altruistic behavior. Stern [22] assumed EPC as a prerequisite of AR. People with AR are likely to feel responsible for their acts and be less likely to perform certain behaviors [34]. That is, people's intentions to choose pro-environmental coffee shops are formulated by attitudes, subjective norms, perceived behavioral control, and PEN [58,59]. Therefore, in order to raise customers' AR and induce them to visit eco-friendly coffee shops, managers have to ensure sustainable management and inform customers about it.

Secondly, SEN has a significant positive effect on PEN. This supports the existing research [3,4, 10,28,32,47,58]. According to Han's [27] study, a significant relationship between social and personal norms and environmentally-friendly purchasing intentions has been demonstrated in the context of hospitality. In addition, the NAM has shown that the SEN and PEN are interrelated and that SEN influences PEN to modify individuals' actual behaviors [23]. Therefore, given SEN's strong influence on PEN, it is important to recognize the problem of the social environment. For example, we now know that SEN has a strong impact on PEN. Therefore, we can insist that increasing PEN is possible by the recognition of individuals/coffee shop managers/social environmental activists of EPC in advance and related to SEN.

Third, the study results show that the effects of factors on pro-environmental behavior intention have a descending order, from PEN to AR to SEN. This fact indicates that SEN's effect on customer behavior is relatively lower than that of PEN, which is actually the most effective factor. Therefore, coffee shop managers should recognize the importance of environmental protection in order to raise the PEN of their customers.

Fourth, we can predict that an increase in coffee shop customers' AR can influence their pro-environmental behavior intention. In other words, AR may generate pro-environmental intentions/behavior among people, which will also lead to responsibility regarding the use of disposable products or plastic and its consequences. Thus, pro-environmental coffee-shop managers should run related campaigns to raise AR of customers and enhance the value of environmental protection in our society. Thereby, the number of coffee-shops under sustainable management can be further expanded to encourage pro-environmental customer.

Lastly, OGI for eco-friendly coffee shops did not moderate the effect of the relationship between SEN and PEN on PCBI. This implies that if SEN and PEN are given, it affects PCBI regardless of coffee shops' OGI. That is, when customers visiting coffee shops recognize EPC, they show PCBI irrespective of the coffee shops' OGI. However, those with low OGI in eco-friendly coffee shops can see that the behavior of eco-friendly customers is higher based on their AR for the environment. Therefore, we can see that the behavior of low-OGI groups can positively enhance PCBI. In other words, the efforts of coffee shop enterprises to protect and preserve the environment are important, but if the consumers try to raise their environmental responsibility, AR for the environment can be improved.

*5.3. Limitations and Future Studies*

Some limitations are as follows: First, future research should consider the impact of national/cultural differences on the proposed theoretical framework. This study was conducted at coffee shops located in Seoul, Republic of Korea. One of the remaining questions is whether the results could be applied to other regional/cultural environments such as Japan, China, or other Asian countries. In addition, a cross-cultural study could be beneficial to investigate PCBI in coffee shops.

Second, most of the moderating effects of OGI were not supported in this study. In this study, OGI was interpreted as having a mediating effect between AR and PCBI only because the low-OGI group

showed a significant link between AR and PCBI and the high-OGI group did not show a significant link between AR and PCBI. Beyond these findings, no study has demonstrated OGI's mediating effect through a chi-square test. The present study's results may or may not indicate a moderating effect because they can be interpreted in various ways depending on different viewpoints. Therefore, in the future, it will be possible to provide more precise theoretical and practical implications through continuous studies on the moderating effects of OGI on the link between EPC and PCBI.

Third, this study did not divide consumers based on their different levels of EPC or the different levels of service they received from the coffee shops they visited. Future studies could investigate different consumer profiles based on different levels of EPC and service and then test the same model to see if any discernible differences between them can be identified.

**Author Contributions:** Conceptualization, S.Y. and T.K.; methodology, S.Y.; formal analysis, S.Y.; investigation, T.K.; resources, T.K.; data curation, T.K.; writing—original draft preparation, S.Y. and T.K.; writing—review and editing, S.Y. and T.K.

**Funding:** This research received no external funding.

**Conflicts of Interest:** The authors declare no conflict of interest.

## Appendix A. Measurement Items

**Environmental problem concern**
I try not to buy from brand of coffee shop that strongly pollute.
When possible, I systematically choose the coffee shop that has the lowest negative impact on the environment.
When I have the choice between two equivalent brand of coffee shop, I always wonder which one pollutes less

**Ascription of responsibility**
I feel jointly responsible for the energy problems.
I feel jointly responsible for the exhaustion of energy sources.
I feel jointly responsible for global warming.
My contribution to the energy problem is not negligible.

**Personal environmental norm**
I feel an obligation to visit an environmentally responsible coffee shops rather than regular coffee shops.
I feel that it is important to make coffee shops environmentally sustainable, reducing the harm to wider environment.
Regardless of what other people do, because of my own values/principles I feel that I should attend an environmentally responsible coffee shop.

**Social environmental norm**
Most people who are important to me think I should visit an environmentally coffee shop.
Most people who are important to me would me to visit with an environmentally responsible coffee shop.
People whose opinions I value would prefer me visit with an environmentally responsible coffee shop.

**Pro-environmental customer behavioral intentions**
I will encourage my friends and relatives to choose an eco-friendly coffee shop.
If someone is looking for a coffee shop, I will advise him/her to choose an environmentally responsible coffee shop.
I will say positive things about an environmentally responsible coffee shop.
I am willing to pay coffees shops with an environmentally responsible coffee shop when deciding on visit coffee shop in the future.
I plan to pay coffee shops by environmentally responsible coffee shop instead of a regular coffee shop in the future. I switch to other brands for ecological reasons.
I would be willing to pay more for an environmentally responsible coffee shop.
To protect the environment, I would expend effort on visiting by an environmentally responsible coffee shop instead of a general coffee shop in the future
To protect the environment, I would be willing to accept any inconvenience (e.g., recycling, reducing water/ energy use, decreasing waste, using the recycling coffee cup) in an environmentally responsible coffee shop.

**Overall green image**
My overall desire to frequent a pro-environmental coffee shop is strong.
My overall impression regarding a pro-environmental coffee shop is important.
My overall desire to visit for the first time a pro-environmental coffee shop is strong.

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
