# Peer review of "What Do Coffee Shop Entrepreneurs Need to Do to Raise Pro-Environmental Customer Behavioral Intentions?"

_sustainability, doi:10.3390/su11092666_

Round 1
Reviewer 1 Report
Overall, this is an interesting expansion of the norm activation model to coffee shop consumption, an important and expanding area of global consumption. My main suggestion is to reduce the number of acronyms throughout the article to improve its readability. There are many acronyms, several of which are never spelled out in the main body of text, making the article unnecessarily difficult to follow. I have several other suggestions for improving the manuscript below. Finally, there are a number of distracting grammatical errors, including missing words in several places. I’ve tried to point several of these out.
Abstract
The sentence starting, “We employed the NAM to test its mediating effect personal environmental norms…” seems to be missing a word. Did the authors mean “its effect mediating personal environmental norms”? Or maybe “test its mediating effect on personal environmental norms”?
The last sentence is too strong of a statement. It should state “may encourage” or something similar.
Introduction
Missing word, p. 1, lines 31-32 “…more people are sustainability-related issues than ever before.” More people are studying sustainability-related issues?
p. 1 “As consumers have recognized the importance of a sustainable green environment, environmental protection has become an essential issue in the consumer market [5].” One could also argue that environmental protection has not been an important issues in the consumer market in many countries globally. If the authors are specifically referring to your study site or another set of locations, this should be specified.
Missing word, p 4 line 91 “3) deepen the NAM by considering the moderating impact of 4)” The moderating impact of what?
p. 3 EPC, PEN, and AR are introduced in the second paragraph but the acronyms are never spelled out in the text before their introduction – the abstract and key words do not count as main text. The same for OGI when it’s introduced in paragraph 2 on p. 6
Again, there are an excessive number of acronyms, which make the article difficult to follow. I recommend selecting one or two to use throughout and spelling out everything else.
Methods
p. 6, “The measurement scales can be found in the literature”
It would be a useful contribution to the literature if the authors included all of their survey questions with the corresponding scales in a table in this article. Some of the items are included in the text and the sources for the remaining items are referenced, but it would be useful for someone that wanted to replicate or expand this study to have all of the questions analyzed in this article included in a table directly in this article. They can be included as an appendix or supplementary information if the authors do not believe they fit as a table within the text
The survey was conducted with a large survey research firm, yet no funding is listed. Did the authors forget to identify a funding source?
Was the survey administered in English or another language? If it was administered in another language, the authors should provide a description of how concepts translated/if they translated well between that language and English.
The authors should provide information on the software used for the statistical analysis, as this is important for replication.
Results
Provide a higher resolution image for figure 1 if possible. Also, it’s difficult to distinguish the arrow colors for supported and unsupported links.
Discussion/conclusions
Lines 440-445 p. 12, This conclusion is not valid unless the authors know of a study supporting that businesses or organizations highlighting their efforts in environmental protection increases peoples’ sense of responsibility for environmental issues. Awareness and responsibility are not the same.
Author Response
Dear Editor and reviewers,
Thank you very much for reviewing our manuscript. We appreciate the opportunity you have afforded us to revise and resubmit. We found your suggestions to be thought-provoking and useful and have worked diligently to improve the manuscript as you suggested. Below, you will find our replies and responses to your constructive comments. Within the responses, red sections denote changes we made to the manuscript itself. Within the manuscript itself, changes are highlighted in red. We hope the changes made are satisfactory to you.

Reviewer 2 Report
A brief summary
This might be an interesting paper which has potential to be published in “Sustainability” Journal but it needs a few improvements.
Pro-environmental customer behavioral intentions have been a subject of growing interest during last years. This paper tries to contribute to this literature and body of research and develops a theoretical framework that explains pro-environmental customer behavioral intentions in the environmentally-responsible coffee shop business.
The Authors broaden the norm activation theory by incorporating the value-belief-norm framework and normative process. They also test the specific mediating and moderating effects.
Nevertheless, this paper needs some important amendments. Therefore, I suggest main recommendations for the Authors’ consideration in order to improve the reviewed paper.
Broad comments
Titles in section 2 (2.1 – 2.4) are incorrect (i.e. multiplication) and must be modified.
In order to identify the common method bias, Harman’s test is recommended.
It is written that coffee shop customers from different regions of the Republic of Korea were surveyed. This suggests that analyzed coffee shops were located in different regions in Korea. However, according to section 5.3., this study was conducted at coffee shops located in Seoul solely. This should be clearly described in section 3.2.
In my opinion, in order to achieve better clarity to the Reader, data concerning indirect effects (Table 2) should be presented in a separate table.
In addition, according to my experience the mediation effects are subjects to very stringent requirements. I would recommend the Authors to test the classic Baron & Kenny approach.
The Authors should be also aware that there is a full and partial mediation effects and it should be noted in the paper.
Figure 1 seems to be unclear and chaotic:
- it presents data resulting from different models, it might be confusing, in my opinion,
- questions related to EPC are presented – in my opinion this figure should show all questions for all variables or none,
- according to section 3.1., PCBI featured eight items but according to Figure 1, there are 3 dimensions of PCBI. This result has been left unmentioned in the main content of the article.
Specific comments
<Line 295> 3.2. Data collection and same characteristics (“same” must be changed)
Table1:
- abbreviation “OGI” should be explained in note 1
- in note 1 it should PCBI instead of PCB (the same situation is in note 1 in figure 1)
<Lines 356-356> “The impact of PEN on PCBI was not assessed; thus, H5 was not supported. The proposed impact of EPC on SEN was assessed.” I assume that it should be “The impact of PEN on PCBI was not significant; thus, H5 was not supported. The proposed impact of EPC on SEN was significant.”
<line 394> “This study several theoretical implications.” Some wording is lacking.
<line 461-465> This paragraph concerns moderating effects but an error occurred and two sentences refers to mediation effect.
Author Response

(The authors gave the same response as above.)
